# A Multi-Component Physiotherapeutic Intervention among Schoolchildren with Myopia: 3D-Based Vision Training Program with Auditory Frequency Entrainment and Electrical Stimulation

**Yu-Kuei Teng** [1,2], **Chi-Wu Chang** [3] and **Shin-Da Lee** [4,5,6,*]

1. School of Nursing, China Medical University, Taichung 406040,Taiwan; tengyk@mail.cmu.edu.tw
2. China Medical University Hospital, Taichung 404332,Taiwan
3. Chung-Shan Medical University Hospital, Taichung 402367, Taiwan; cshy646@csh.org.tw
4. Department of Physical Therapy, Asia University, Taichung 41354, Taiwan
5. School of Rehabilitation Medicine, Weifang Medical University, Weifang 261053, China
6. Department of Physical Therapy, China Medical University, Taichung 406040,Taiwan
* Correspondence: shinda@mail.cmu.edu.tw

**Abstract:** Purpose. This study evaluated whether 3D-based vision training (VT) with visual cortex-activated auditory frequency entrainment and bilateral orbital electrical stimulation (ES) could prevent the progression of myopia among schoolchildren. Methods. In this two-stage, randomized, crossover, single-blind study, pre- and post-logMAR visual acuity and refractive error from 27 schoolchildren with myopia ($\leq -0.50$ D) were evaluated among four groups: (1) sham control with no VT, frequency following response (FFR), or ES (control group); (2) 3D-based VT only (VT group); (3) VT with FFR generated through visual cortex-activated auditory entrainment (VT-FFR group); and (4) VT with FFR and bilateral orbital ES (VT-FFR-ES group). In stage 1, the intervention was administered for 30 min to all groups using a randomized crossover design. In stage 2, the intervention was administered for 30 min/day, 3 days a week, for 4 weeks to evaluate the effectiveness of intervention. Results. Compared with the pre-test, post-test logMAR visual acuity after a single intervention was not significantly different in control and VT groups, but significantly improved in the VT-FFR ($-0.08 \pm 0.11$) and VT-FFR-ES groups ($-0.13 \pm 0.14$). Compared with the pre-test, post-test refractive error by spherical equivalent in VT-FFR-ES group for the 4-week intervention was significantly (<0.001) improved (0.21 D) compared with the control group ($-0.1$ D). Conclusions. The multicomponent physiotherapeutic intervention of 3D-based VT with auditory FFR and bilateral orbital ES can inhibit the progression of myopia. This intervention can be used as a noninvasive physiotherapeutic approach to prevent or reduce the severity of myopia.

**Keywords:** binaural; children; myopic; near-sightedness; visual entrainment

## 1. Introduction

Myopia, also known as near-sightedness, is one of the most common types of refractive errors and can lead to blindness. It is a worldwide public health issue. Myopia (spherical equivalent > $-0.5$ D) and high myopia (spherical equivalent > $-5.0$ D) were estimated to affect 22.9% (1406 million) and 2.7% (163 million) of the world's population, respectively, in 2000, and these values are projected to increase to 50% (4758 million) and 9.8% (938 tmillion), respectively, by 2050 [1]. In Taiwan, early-onset myopia ($\leq$7 years of age) may increase the risk of high myopia among schoolchildren [2,3]. High myopia is associated with complications such as cataracts, glaucoma, retinal detachment, myopic macular degeneration, visual impairment, and blindness [1,4–6]. The World Health Organization (WHO) has reported that uncorrected refractive errors are the main reason for 42% of visual impairments and 3% of blindness [7]. However, 80% of visual impairments are avoidable, and even a 50% reduction in the rate of developing myopia can decrease the prevalence of

high myopia by as much as 90% [7,8]. Eliminating the main causes of all preventable and treatable blindness and improving vision health are paramount [9].

The proposed mechanisms underlying myopia development include excessive accommodation and uncoordinated eye growth due to the response of retinal signals to sustained extended near work [10]. The current common therapeutic approaches to myopia include the use of glasses or contact lenses, laser eye surgery, and atropine eyedrops [11]. Alternative approaches to prevent and delay the progression of myopia include avoiding using eyes at close distances for a long period, increasing outdoor time [12,13], acupressure or electroacupuncture [14–16], and eye exercise training [17–19].

Eye exercises are often used in physiotherapeutic optometric vision therapy, which establishes effective ways of using the eyes as well as muscle relaxation techniques, biofeedback, eye patches, or eye massages, alone or in combination, to improve visual clarity [18,19]. However, few studies have comprehensively evaluated a 3D-based vision training (VT) program to improve visual clarity. Auditory frequency following responses (FFRs), generated by periodic or nearly periodic auditory stimuli at low frequencies, is a powerful neural processing tool of the visual cortex and affects the optic and visual nerves [20]. Electrical stimulation (ES) or electrical acupuncture/acupoint stimulation is a safe and effective technique that may have therapeutic applications in controlling myopia [16,21–24].

Binaural differences in the alpha-band (8–14 Hz) stimulus recognize the visual target and causes the visual stimulation of resonance rhythms [25]. However, the effect of 3D-based VT plus FFR generated by auditory binaural frequency entrainment remains unknown.

ES is commonly used in physical therapy to relax hypertonic muscles, reduce muscle spasms, and improve muscle discomfort symptoms [26]. This response is effected through the excitation–contraction–relaxation cycle in muscles by mediating $Ca^{2+}$ levels [27]. ES was reported to reduce tetanic tension in an animal model [28]. ES is often applied directly to acupoints or motor points in the muscle. Bilateral orbital Taiyang (EX-HN5) acupoints have been reported to prevent myopia in animal studies [16], but no human trials have been conducted.

In the present study, we applied 3D-based VT, FFR generated by visual cortex–activated auditory binaural frequency entrainment, and ES to evaluate the best observed outcome intervention and examine the effectiveness of multimodal interventions in preventing the progression of myopia among Taiwanese schoolchildren. Our findings may provide useful information for the promotion of visual health and prevention of deterioration for schoolchildren and their parents.

## 2. Materials and Methods

All procedures were reviewed and approved by the Institutional Review Board of China Medical University Hospital (CMUH107-REC1_124), and all tests were conducted in accordance with the tenets of the Declaration of Helsinki. After children and their parents were provided a description of the study aims and procedures, written informed consent was obtained from the parents of all participating children. Our study was recorded in the U.S. National Library of Medicine registry (NCT04017234). All ongoing and related trials for this intervention have been registered.

### 2.1. Participants

We recruited 27 children aged 8–12 years from an elementary school in central Taiwan. The following inclusion criteria were applied: (a) presence of myopia (defined as a spherical equivalent of at least one $\leq -0.50$ D); (b) non-use of a cycloplegic agent or ophthalmic drug (such as cyclogel or atropine) for the past $\geq 7$ days; (c) non-use of orthokeratology lens currently or in the past $\geq 7$ days; and (d) ability to answer the questionnaire. The exclusion criteria were active or recent eye trauma or irreversible eye diseases affecting visual acuity.

*2.2. Vision Training*

　　　The virtual 3D-based VT system was designed by our team to create a 3D environment (Figure 1) with a deep enough depth of field, and the eyes were guided to trace a light or contrast color spot that could move near or far away, right or left, up or down, clockwise or counterclockwise, and fast or slow. This system was designed to reduce the spasm of the oblique and rectus muscles and to relax the ciliary muscle to focus on distant objects.

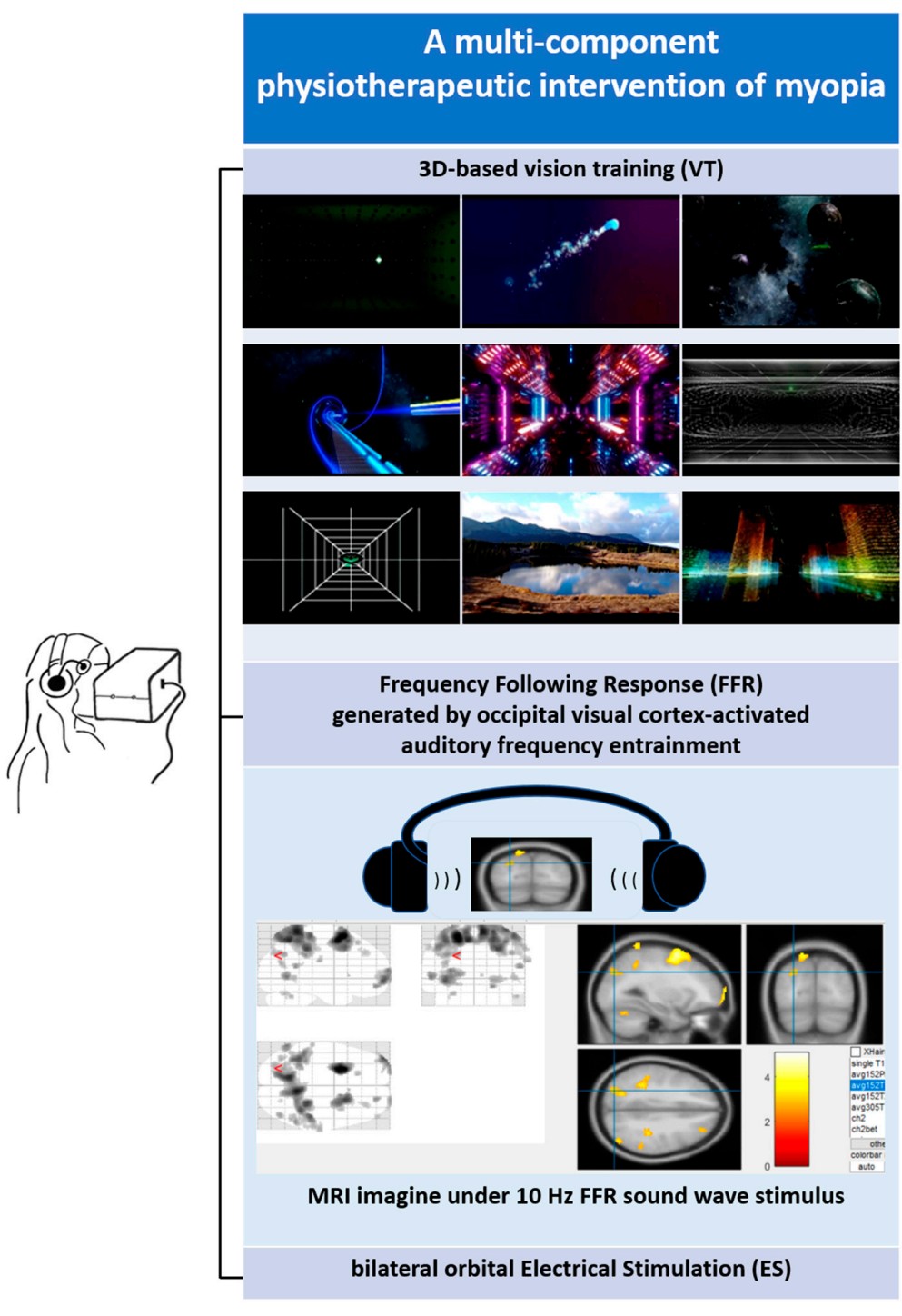

**Figure 1.** A multi-component physiotherapeutic intervention of myopia prevention.

### 2.3. FFR

FFR was generated by various difference auditory binaural frequency entrainment (8–14 Hz) with an activated visual cortex, as shown in the magnetic resonance imaging (MRI) findings (Figure 1). In the current study, a 3D-based VT system with auditory binaural frequency (8–14 Hz) entrainment was applied during a single VT-FFR and VR-FFR-ES intervention and chronic VR-FFR-ES intervention.

### 2.4. Bilateral Orbital ES

ES with a direct current (2–100 Hz, 1–12 mA) was applied to the bilateral orbital area approximately one fingerbreadth posterior to the midpoint between the lateral end of the eyebrow and the outer canthus for 30 min. The ES intensity was adjusted by the participants themselves in increments of 1 mA to an acceptable level.

### 2.5. Study Design

A two-stage, randomized, crossover, single-blind intervention design was used in this study. In the first stage, a single 30 min intervention was used in each group, with a randomized crossover design to investigate the effectiveness of a 3D-based VT program with or without FFR and ES. In the second stage, a chronic intervention for 30 min/day, 3 days a week, for 4 weeks with a randomized crossover design was applied to evaluate the effectiveness of the best multicomponent physiotherapeutic intervention of VT from the first stage. Figure 2 presents a flow diagram of the study design and participant selection. The 3D-based VT program was performed at a primary school. Each intervention session lasted for approximately 30 min. One researcher allocated participants to groups using a randomized crossover design in stage 1 and stage 2. The other experimenters were blinded to participant allocation.

### 2.6. Outcome Evaluation

Both before and after the intervention (pre-test and post-test, respectively), visual acuity in stage 1 and spherical equivalents in stage 2 were evaluated by an optometrist who was blinded to the participant allocation. The visual acuity was determined with the Snellen chart of a decimal visual record at a viewing distance of 6 m. For statistical analysis, we converted the values of the decimal visual acuity chart to the log of the minimum angle of resolution (logMAR) visual acuity. Spherical equivalents were evaluated using an auto-refractometer, and three readings were obtained and averaged for each eye in all participants for further analysis (Topcon KR-8100, Tokyo, Japan).

### 2.7. Protocol of Stage 1

This study applied a randomized crossover design (which was randomized through the sequential arrangement of each intervention, including four treatments, controlling for the effects of VT, FFR, and ES) to investigate the effectiveness of the 3D-based VT program with or without FFR generated by visual cortex-activated auditory frequency entrainment and bilateral orbital ES. A total of 27 school children with myopia were randomized through sequential arrangement into four groups: (1) sham control with no VT, FFR, or ES (control group); (2) 3D-based VT only (VT group); (3) VT with FFR (VT-FFR group); and (4) VT with FFR and ES (VT-FFR-ES group). Each participant completed all three interventions and the control test (30 min per intervention) with a washout period of at least 3 days between interventions. For each intervention, the pre-test and post-test visual acuity was measured using the Snellen chart, and the best effective multicomponent physiotherapeutic single intervention of myopic improvement was identified.

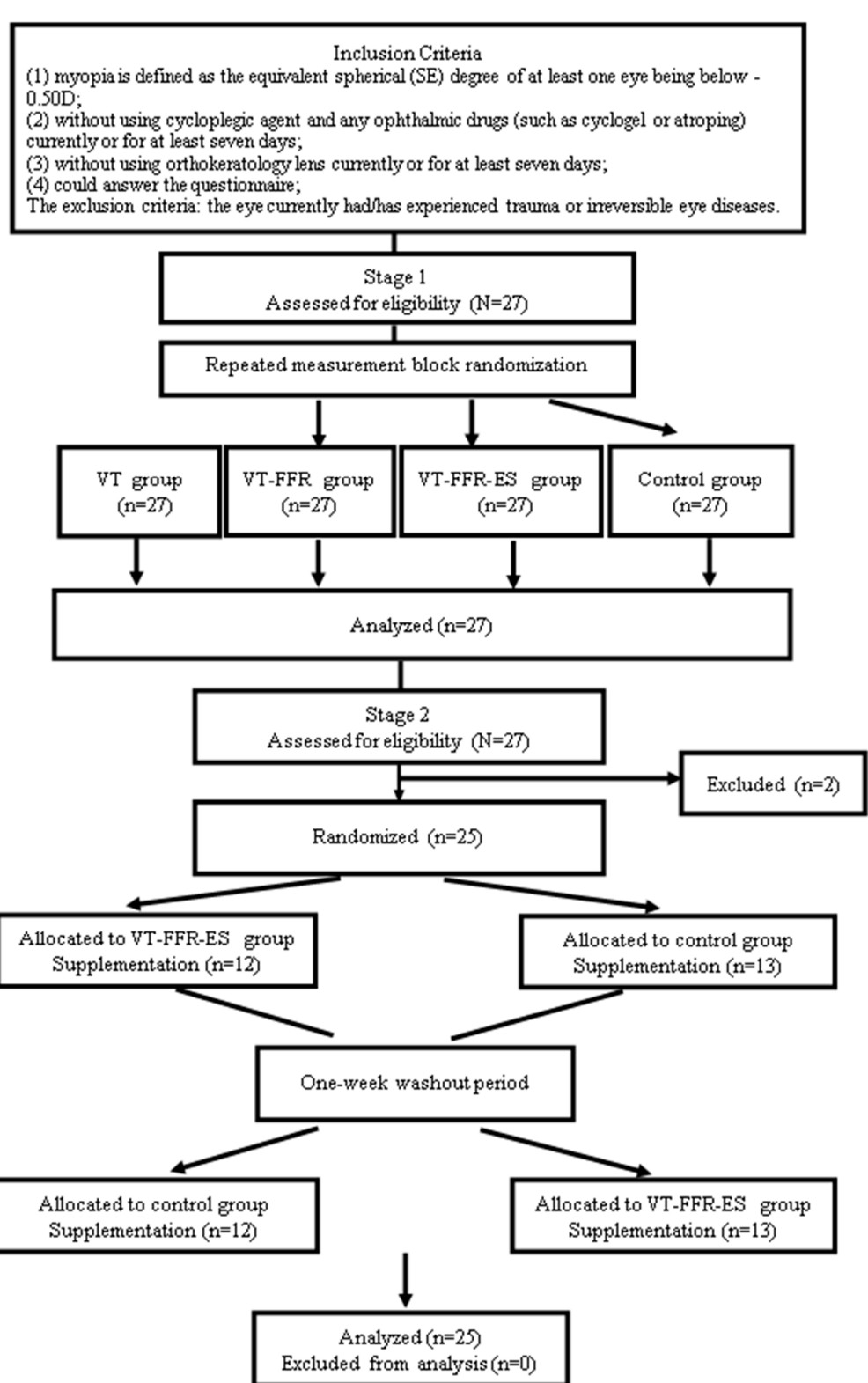

**Figure 2.** Flow chart depicting the total number of participants and how the final study enrollment was achieved. VT: Vision training; VT-FFR: Vision training and frequency following response; VT-FFR-ES: Vision training, frequency following response and electrical stimulation; Control group: No function group of VT-FFR-ES.

*2.8. Protocol of Stage 2*

The participants were divided into control and VT-FFR-ES groups (n = 13 and 12, respectively) and underwent a longer-term intervention (30 min/day, 3 days a week for 4 weeks) in a crossover design to evaluate the effectiveness of the best multicomponent physiotherapeutic intervention, VT-FFR-ES, which was chosen from the outcome of stage 1. Before the crossover, a 1-week washout period was applied, and this duration was determined by the restoration of visual acuity after using cycloplegic drugs (atropine), which are effective in suppressing the progression of myopia. Participants did not receive any other VT or myopia prevention intervention or treatment throughout the study (both stages).

*2.9. Statistical Analysis*

The outcome of visual acuity was used to calculate the sample size of the study. To reach a power of 0.8, a two-sided significance level of 0.05 was required if the difference between the interventions was 0.5; assuming that the within-subject standard deviation of the outcome was 0.8, a sample size of 23 schoolchildren was estimated. All analyses were conducted using SPSS 22.0 (IBM, Chicago, IL, USA). All visual acuity and spherical equivalent outcomes of the eyes were statistically compared with those of the fellow eyes within the same intervention using a paired-sample *t* test. A repeated-measures analysis of variance (ANOVA) was used to compare the effect on outcomes, such as logMAR visual acuity, between control and the three interventions. Data are expressed as the mean $\pm$ standard derivation. The level of statistical significance was set at $p < 0.05$.

**3. Results**

The 27 participants were arranged by randomized crossover design for each intervention in stage 1; for personal reasons, two participants withdrew in stage 2. The participants' clinic demographic data are presented in Table 1. The age range of the participants was 10–12 years old, 40.7% of the participants were boys, and 80.19% of the participants had experienced no eye discomfort in the previous 3 months.

**Table 1.** Demographic data of the participants.

| Variables | | Frequency | Percentage |
|---|---|---|---|
| Gender | | | |
| | Male | 11 | 40.7 |
| | Female | 16 | 59.3 |
| Age (years) | | | |
| | 10 | 9 | 33.3 |
| | 11 | 12 | 44.4 |
| | 12 | 6 | 22.2 |
| Myopia status of father | | | |
| | No | 13 | 48.2 |
| | Yes | 14 | 51.8 |
| Myopia status of mother | | | |
| | No | 15 | 55.6 |
| | Yes | 12 | 44.4 |
| Education of father | | | |
| | $\leq$12 years | 11 | 40.7 |
| | >12 years | 16 | 59.3 |
| Education of mother | | | |
| | $\leq$12 years | 9 | 33.3 |
| | >12 years | 18 | 66.7 |
| Outside activity during the previous 3 month | | | |
| | No | 3 | 11.1 |
| | Yes | 24 | 88.9 |

Table 2 summarizes the postinterventional changes in logMAR visual acuity. Compared with the values before the intervention, the postinterventional logMAR visual acuity was significantly different in the VT-FFR ($0.63 \pm 0.36$ vs. $0.56 \pm 0.32$, $p < 0.001$) and VT-FFR-ES groups ($0.60 \pm 0.34$ vs. $0.46 \pm 0.28$, $p < 0.001$). Thus, the VT-FFR-ES intervention was identified as having the best observed outcome for logMAR visual acuity (i.e., the greatest decrease in logMAR visual acuity) (Table 2). Figure 3 presents an identity plot for logMAR visual acuity in the No function group and VT-FFR-ES group after a single intervention.

**Table 2.** The logMAR visual acuity of pre–post-evaluation of single intervention (n = 54 eyes).

| Intervention | Before | After | Mean Difference | $p$ [a] |
|---|---|---|---|---|
| Control | $0.60 \pm 0.33$ | $0.58 \pm 0.33$ | $-0.02 \pm 0.08$ [b,c] | 0.08 |
| VT | $0.62 \pm 0.36$ | $0.54 \pm 0.33$ | $-0.07 \pm 0.43$ | 0.22 |
| VT-FFR | $0.63 \pm 0.36$ | $0.56 \pm 0.32$ | $-0.08 \pm 0.11$ [d] | <0.001 |
| VT-FFR-ES | $0.60 \pm 0.34$ | $0.46 \pm 0.28$ | $-0.13 \pm 0.14$ | <0.001 |

Control: no function of VT-FFR-ES; VT: vision training; FFR: frequency following response; ES: electrical stimulation. [a] paired *t*-test [b] Control > VT-FFR [c] Control > VT-FFR-ES [d] VT-FFR >VT-FFR-ES by repeated-measures ANOVA with least significant difference post hoc test.

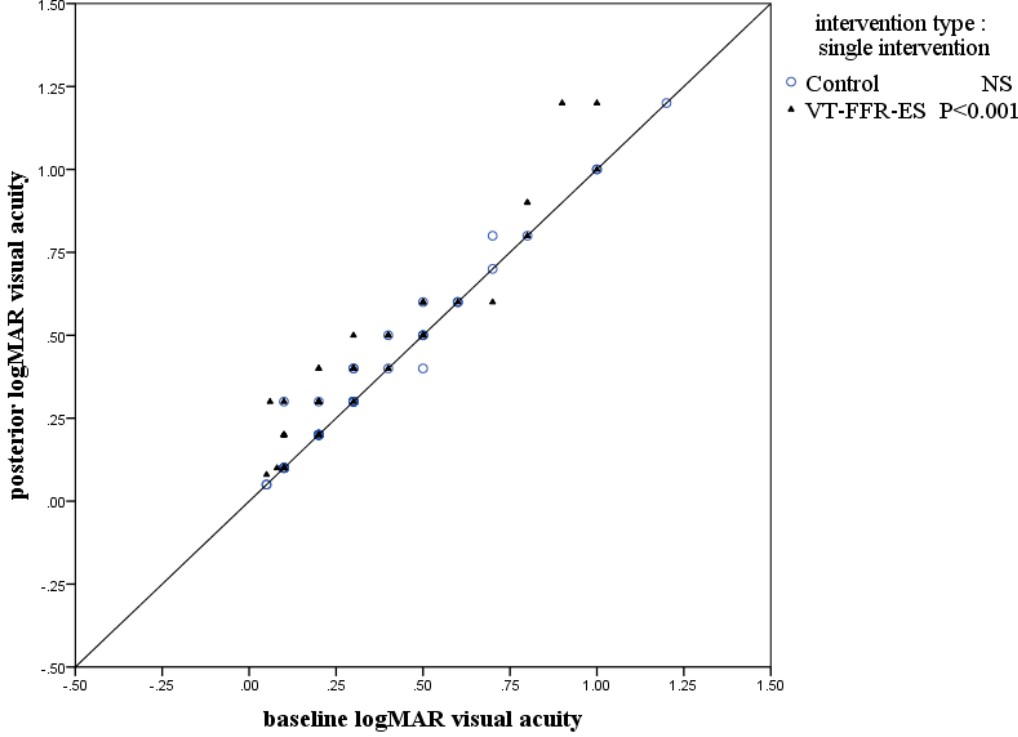

**Figure 3.** The identity plot presents the distribution of pre- and post-individual logMAR visual acuity between the single Control and VT-FFR-ES intervention types. The upper half of the identity line means a worse visual acuity than the baseline; the lower half of the identity line means a better visual acuity than the baseline. VT-FFR-ES: vision training, frequency following response and electrical stimulation; Control: no function group of VT-FFR-ES. Paired *t*-test NS, non-significant and $p < 0.001$.

In the chronic intervention, the average spherical equivalents in the VT-FFR-ES and Control groups were $-2.59 \pm 1.88$ D and $-2.56 \pm 1.79$ D, respectively, in the pre-test, and $-2.38 \pm 1.72$ D and $-2.66 \pm 1.91$ D, respectively ($p < 0.001$, Table 3), in the post-test. Figure 4 presents an identity plot for pre- and post-test spherical equivalent values after 4 weeks of chronic intervention.

**Table 3.** Refractive error between chronic Control and VT-FFR-ES interventions (n = 50 eyes in each group).

| | Control Group | | | VT-FFR-ES | | | |
|---|---|---|---|---|---|---|---|
| | **Before** | **After** | **Change in Refraction** | **Before** | **After** | **Change in Refraction** | ***p* \*** |
| Spherical equivalent (D) | $-2.56 \pm 1.79$ | $-2.66 \pm 1.91$ | $-0.10 \pm 0.26$ | $-2.59 \pm 1.88$ | $-2.38 \pm 1.72$ | $0.21 \pm 0.32$ | <0.001 |

Data shown are the mean ± SD. VT: vision training; FFR: frequency following response; ES: electrical stimulation; D: diopters. VT-FFR-ES: vision training, frequency following response, and electrical stimulation; Control group: no function group of VT-FFR-ES. * *p*-Values derived from paired *t*-tests of the mean differences between the Control group and VT-FFR-ES intervention. Paired *t*-test NS, non-significant and *p* < 0.001.

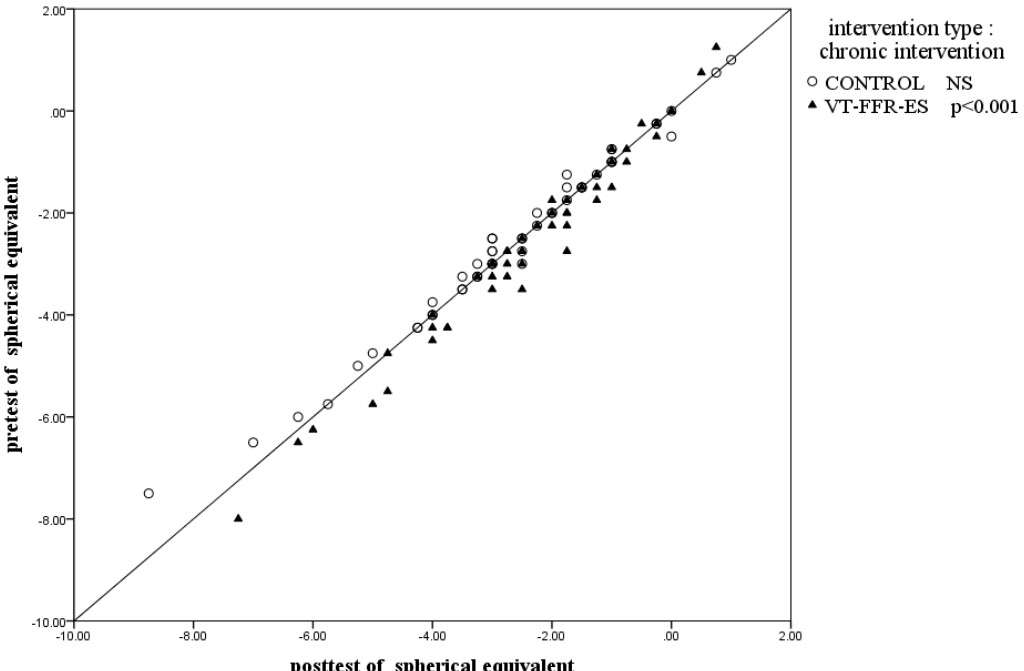

**Figure 4.** Identity plot between pre- and post-test spherical equivalents after chronic VT-FFR-ES intervention.

Identity plot between pre- and post-test spherical equivalents between the Control group and vision training with frequency following response and electrical stimulation group (VT-FFR-ES) for 4 weeks.

### 4. Discussion

This study had two main findings. Firstly, a single multicomponent physiotherapeutic intervention of a 3D-based VT program with FFR generated by auditory frequency entrainment, and bilateral orbital ES for 30 min, improved visual acuity among schoolchildren compared with a control sham intervention or 3D-based VT only. Secondly, the 4-week application of the multicomponent physiotherapeutic intervention VT-FFR-ES (30 min/day, 3 days a week) significantly improved refractive error, as measured using spherical equivalents, compared with the control. Myopia is a common problem in schoolchildren [1], and efficacious public health interventions are urgently warranted to prevent or delay the progression of myopia in schoolchildren. The most effective strategy for controlling myopia has yet to be identified [29]. Nonpharmacological or noninvasive ES interventions for treating myopia have been inconclusive. A meta-analysis between outdoor time and reducing the risk of developing myopia, a notable protective effect of outdoor time, was found for incident myopia in clinical trials and longitudinal cohort studies, and prevalent

myopia in cross-sectional studies (reducing the risk to 46%, 43%, and 4%, respectively). The pooled results of outdoor time in clinical trials showed that there was a reduced myopic shift in both myopes and non-myopes (−0.30 D), and compared with a control group (weighted mean difference = −0.30 D, 95% CI = −0.18 to −0.41), over a 3-year follow-up period. However, only in among myopes, no dose–response relationship was found in the analysis between time outdoors and myopic progression ($R^2$= 0.00064). Increased outdoor time is effective in preventing the onset of myopia and slowing the myopic shift. Unexpectedly, outdoor time was not an effective strategy in slowing progression in the eyes that were already myopic [30]. Therefore, VT is considered an eye exercise training for preventing from myopic progression, which is responsible for changing the focus of the eyes. Another nonpharmacological therapy that was tested was VT with the simultaneous use of soft lenses in 14–22-year-old patients with myopia [18]. The results revealed that VT significantly affected shorter distant viewing times; however, this outcome of the evaluation differed from ours (visual acuity and spherical equivalent). One study observed that daily VT-based eye exercises for 6 weeks significantly improved visual acuity in 12–15-year-old girls with myopia [31]. This result partially supported our result for a 4-week intervention of VT-FFR-ES for 30 min/day, 3 days a week. An information and communication technology (ICT)-based VT program involving a head-mounted wearable device in young adults (22.8 ± 3.8) stimulated the contraction and relaxation of ciliary muscles of the eye and improved refractive errors by 0.44 ± 0.35 D in a VT group compared with a comparative group [32]. VT involves the contraction and relaxation of the ocular ciliary muscles, which, in turn, improves vision. In the present study, because the 3D-based VT alone did not improve visual acuity, we assumed that it was not effective enough, and that applying the multicomponent intervention of VT-FFR-ES is better for controlling the progression of myopia.

Some links between auditory-induced FFR and the visual system can be found. FFR is a footprint of early auditory nerve processing and is related to the optic nerve [20]. The interactive effect of binaural tones of different intensities was reported on visual perception [33]. The current study provided MRI evidence of visual cortex activation through FFR generated by various difference auditory binaural frequency (8–14 Hz) entrainment regimes. This is the first report to demonstrate an activated visual cortex by FFR or binaural beats. In our study, the significant effect of visual acuity is the first evidence of a single intervention of a 3D-based VT program with auditory frequency entrainment. Unexpectedly, we found that the single 3D-audiovisual intervention, a 3D-based VT program with auditory 8–14 Hz frequency entrainment, could improve visual acuity in schoolchildren with myopia.

ES is a promising, safe, and effective therapeutic modality for improving vision [21]. Electrical acupuncture is commonly used to control myopia in children and adolescents [22,23]. Several studies have evaluated the effectiveness of electrical acupuncture stimulation at the Taiyang acupoint for myopia in an animal model with lens-induced myopia [16,24], and reported that ES improved neuromuscular nodulation (such as retinal GABA) and increased resting-state functional connectivity in visual cortices in lens-induced myopic eyes, but did not influence refractive error or axial length [16]. The Taiyang acupoint is often selected for treating myopia in traditional Chinese medicine, because it is located around the bilateral orbital area. Some mixed acupoint stimulations for myopia therapy may clear the meridians, improve chorioretinal microcirculation, enhance cerebral circulation [34,35], and activate the visual cortex [36]. This is the first report to demonstrate that a single 30-min session of VT-FFR-ES can effectively improve visual acuity, and that a 12-session program of VT-FFR-ES over 1 month can effectively prevent myopia among schoolchildren.

The main strength of the present study is the use of a randomized crossover design to identify the best intervention for improving visual acuity in children with myopia. However, this study has some limitations. First, our participants were schoolchildren from a single school in central Taiwan and may be susceptible to the development of myopia at their age, thereby precluding the generalizability of our findings to other target populations.

Second, the 1-month follow-up period precluded insights into the long-term effects of the multicomponent intervention. Finally, we cannot confirm the intervention effects on the elongation of the eye (i.e., measurements of axial length) and did not assess the rebound effects during the short follow-up period.

## 5. Conclusions

The multicomponent physiotherapeutic intervention of a 3D-based VT program with auditory frequency entrainment and bilateral orbital ES with relaxing the extraocular muscle can effectively prevent or inhibit myopia progression, as demonstrated by the improved visual acuity and refractive error via visual cortex activation and extraocular muscle relaxation among schoolchildren. Further research is required to determine the long-term efficacy of this intervention on myopia prevention and changes in specific eye indicators, and verify the results in other populations and settings.

**Author Contributions:** Conceptualization, Y.-K.T. and S.-D.L.; methodology, Y.-K.T., C.-W.C. and S.-D.L.; validation, Y.-K.T., C.-W.C. and S.-D.L.; formal analysis, Y.-K.T. and S.-D.L.; investigation, Y.-K.T.; resources, S.-D.L.; data curation, Y.-K.T. and S.-D.L.; writing—original draft preparation, Y.-K.T.; writing—review and editing, S.-D.L.; supervision, S.-D.L.; project administration, Y.-K.T.; funding acquisition, S.-D.L. All authors have read and agreed to the published version of the manuscript.

**Funding:** This study was partially supported by 107-2314-B-468-002-MY3 from the Ministry of Science and Technology Taiwan as well as Weifang Medical University. The funders were not associated with the design, data search, data collection, synthesis, or publication decision.

**Institutional Review Board Statement:** All procedures were reviewed and approved by the Institutional Review Board of China Medical University Hospital (CMUH107-REC1_124), and all tests were conducted in accordance with the tenets of the Declaration of Helsinki.

**Informed Consent Statement:** Informed consent was obtained from the parents of all participating children involved in the study.

**Data Availability Statement:** The data presented in the study are available on request from the corresponding author.

**Acknowledgments:** We gratefully acknowledge Yue-Mei Chen and Kang-Ming Lin for developing the virtual 3D-based VT film system. Our gratitude goes to Michael Hsu for producing the auditory frequency entrainment and Cheng-Yi Chen for analyzing the MRI scans. We thank MDPI editing for proofreading. Finally, we thank the children and their parents who willingly and cooperatively participated in the study.

**Conflicts of Interest:** No potential conflicts of interest are reported by the authors.

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
