# Peer review of "A Multi-Component Physiotherapeutic Intervention among Schoolchildren with Myopia: 3D-Based Vision Training Program with Auditory Frequency Entrainment and Electrical Stimulation"

_applsci, doi:10.3390/app12010201_

Round 1

Reviewer 1 Report

This study uses a novel vision therapy technique in a short course treatment trial to try and determine if there is any effect of treatment.

Line 18 mentions NF, but does not say what NF means. I believe, looking at line 201, NF means "no function." Perhaps is would be better to refer to this group as the "no treatment" or control group because "no function" has a different meaning in English.

This paper suggests that myopia is caused by hypertonic muscles, or stressed muscles. It is not at all clear that is the case. In fact, myopia tends to be caused by enlongated axial length suggesting some interactive mechanism between the retina and the sclera. This is why follow up periods for myopia studies are usually one year minimum and up to 3 years or more. In reading the methods, it appears the follow up period is 4 weeks. That suggests that this method is not addressing the fundamental cause of myopia but rather is inducing some type of relaxation of accommodation. It makes me wonder if all of this elaborate electronic instrument treatment could be replaced by simply sending the child outside bright sunshine to play for a half hour. That treatment has been shown in multi-year follow up clinical trials to be effective.

Lines 138-139. It is not clear to me if refractive error was assessed using cycloplegia or not. Please clarify. It would be best assess refractive error using cycloplegia and to include axial length measures.

Lines 164. It is not clear to me what the "cycloplegic drugs" here are referring to. Are these children also on atropine treatment or are the authors referring to the cycloplegics used to assess refractive error?

Line 305. It is not at all clear to me that this treatment is preventing or retarding myopia. This study suggests that the treatment might help to relax a hypertonic accommodative system, and that might be helpful, I don't know. But from the short term study design I think suggesting the treatment prevents or retards myopia is overstating the results.

Author Response

Comments and Suggestions for Authors

Reviewer 1: This study uses a novel vision therapy technique in a short course treatment trial to try and determine if there is any effect of treatment.

Reply: Thanks for positive comment for a novel vision therapy technique.

Reviewer 1: Line 18 mentions NF, but does not say what NF means. I believe, looking at line 201, NF means "no function." Perhaps is would be better to refer to this group as the "no treatment" or control group because "no function" has a different meaning in English.

Reply: Thank you for good suggestion. “NF group” has been changed into “control group”.

Reviewer 1: This paper suggests that myopia is caused by hypertonic muscles, or stressed muscles. It is not at all clear that is the case. In fact, myopia tends to be caused by enlongated axial length suggesting some interactive mechanism between the retina and the sclera. This is why follow up periods for myopia studies are usually one year minimum and up to 3 years or more. In reading the methods, it appears the follow up period is 4 weeks. That suggests that this method is not addressing the fundamental cause of myopia but rather is inducing some type of relaxation of accommodation. It makes me wonder if all of this elaborate electronic instrument treatment could be replaced by simply sending the child outside bright sunshine to play for a half hour. That treatment has been shown in multi-year follow up clinical trials to be effective.

Reply: Thank you for your thoughtful comment and suggestion. We surprisingly found and report the effects of visual acuity within one month in this study and we will process multi-year follow up clinical trials to observe axial length. The multicomponent physiotherapeutic intervention of a 3D-based VT program with auditory frequency entrainment and bilateral orbital ES with relaxing the extra-ocular muscle can effectively prevent or retard inhibit myopia progression, as demon-strated by the improved visual acuity and refractive error via visual cortex activation and extraocular muscle relaxation among schoolchildren. According to a meta-analysis of Xiong et al. (2017), it was a significant association between outdoor time and reducing the risk of developing myopia in incident myopia in clinical trials and longitudinal cohort studies and prevalent myopia in cross-sectional studies. But paradoxically, outdoor time was not effective in slowing progression in the eyes that were already myopic. The 3D-based VT alone did not improve visual acuity in our study, we assumed that it is not effective enough and that applying the multicomponent intervention of VT-FFR-ES is better for controlling myopia progression.

We added the description in the section of the discussion as follows. (Please see lines 254-264.)

“A meta-analysis between outdoor ………. Increased outdoors time is effective in preventing the onset of myopia and slowing the myopic shift. Unexpectedly, outdoor time was not an effective strategy in slowing progression in the eyes that were already myopic.”

Reference: Xiong, S.; Sankaridurg, P.; Naduvilath, T.; Zang, J.; Zou, H.; Zhu, J.; Lv, M.; He, X.; Xu, X. Time spent in outdoor activities in relation to myopia prevention and control: a meta-analysis and systematic review. Acta Ophthalmol 2017, 95, 551-566, doi:10.1111/aos.13403.

Reviewer 1: Lines 138-139. It is not clear to me if refractive error was assessed using cycloplegia or not. Please clarify. It would be best assess refractive error using cycloplegia and to include axial length measures.

Lines 164. It is not clear to me what the "cycloplegic drugs" here are referring to. Are these children also on atropine treatment or are the authors referring to the cycloplegics used to assess refractive error?

Reply: Thank you for good suggestion. The children and their parents don’t want using any drug during the study. Therefore, we assessed refractive error without using cycloplegia to control accommodative problems, because the time of examination of visual acuity in stage 1 and spherical equivalent in stage 2 is before class in the morning to children, who get enough rest during sleeping. We used crossover design to control confounding issue (such as homogeneity of the characteristics of the participants that affect the outcome) among participates. We add the limitation without measuring axial length during the short follow-up period in the discussion. (Please see page 10 lines 308-314, the section of discussion).

Reviewer 1: Line 305. It is not at all clear to me that this treatment is preventing or retarding myopia. This study suggests that the treatment might help to relax a hypertonic accommodative system, and that might be helpful, I don't know. But from the short term study design I think suggesting the treatment prevents or retards myopia is overstating the results.

Reply: We adopted the comments and explained conservatively in the statement of the results.

“In the present study, because the 3D-based VT alone did not improve visual acuity, we assumed that it is not effective enough. Therefore, based on effectiveness of single VT-FFR-ES, we apply the multicomponent intervention of VT-FFR-ES for looking for better approach to control myopia progression.” (Please see page 9 lines 278-280, the section of discussion).

Reviewer 2 Report

General Comments

In this paper, the authors evaluated whether 3D-based vision training with visual cortex activated auditory frequency entrainment and bilateral orbital electrical stimulation can prevent myopia progression among schoolchildren.

The study is of great interest for the increase in this ametropie in the general population but, especially, in children.

Attached suggestion of changes that should be addressed to improve the manuscript.

Specify Comments

# 1 Participants: Within the exclusion criteria, were accommodative problems taken into account?

Couldn't the number of children be expanded? for the sample to be significant, the number of children in the study should be greater.       

# 2 Results. Explain why 2 children attended the study.

One of the most important limitations is not having long-term results. The problem with myopia is the increase over time. It would be interesting to know if the decrease in myopia is maintained over time.

The study is interesting and I encourage the authors to continue with this line of research.

Author Response

Reviewer 2: Comments and Suggestions for Authors

Reviewer 2: General Comments In this paper, the authors evaluated whether 3D-based vision training with visual cortex activated auditory frequency entrainment and bilateral orbital electrical stimulation can prevent myopia progression among schoolchildren.

The study is of great interest for the increase in this ametropie in the general population but, especially, in children.

Attached suggestion of changes that should be addressed to improve the manuscript.

Reply: Thank you very much for the encouragement. 

Reviewer 2: Specify Comments

Reviewer 2: # 1 Participants: Within the exclusion criteria, were accommodative problems taken into account?

Couldn't the number of children be expanded? for the sample to be significant, the number of children in the study should be greater.  

Reply: Thank you for the reminder.   We did not list accommodative problems for exclusion criteria, because the time of examination of visual acuity in stage 1 and spherical equivalent in stage 2 is before class in the morning to children, who get enough rest during sleeping. The number of the study was calculated and the effect size reached over 0.8.     

Reviewer 2: # 2 Results. Explain why 2 children attended the study.

One of the most important limitations is not having long-term results. The problem with myopia is the increase over time. It would be interesting to know if the decrease in myopia is maintained over time.

The study is interesting and I encourage the authors to continue with this line of research.

Reply: We are thankful for your inspirational recommendation. Yes, a long-term follow up will be observed to see the effect of myopia prevention and axial length. We recruited 27 children by estimating the sample size of the study by power and the difference of outcome between the interventions. We also calculate the effect size getting over 0.8.